# ESD: Expected Squared Difference as a Tuning-Free Trainable Calibration Measure

**Hee Suk Yoon**[1][*]    **Joshua Tian Jin Tee**[1][*]    **Eunseop Yoon**[1]    **Sunjae Yoon**[1]
**Gwangsu Kim**[1]    **Yingzhen Li**[2]    **Chang D. Yoo**[1][†]
[1]Korea Advanced Institute of Science and Technology (KAIST)    [2]Imperial College London
{hskyoon,joshuateetj,esyoon97,sunjae.yoon}@kaist.ac.kr
s88012@gmail.com   yingzhen.li@imperial.ac.uk   cd_yoo@kaist.ac.kr

## Abstract

Studies have shown that modern neural networks tend to be poorly calibrated due to over-confident predictions. Traditionally, post-processing methods have been used to calibrate the model after training. In recent years, various trainable calibration measures have been proposed to incorporate them directly into the training process. However, these methods all incorporate internal hyperparameters, and the performance of these calibration objectives relies on tuning these hyperparameters, incurring more computational costs as the size of neural networks and datasets become larger. As such, we present Expected Squared Difference (ESD), a tuning-free (i.e., hyperparameter-free) trainable calibration objective loss, where we view the calibration error from the perspective of the squared difference between the two expectations. With extensive experiments on several architectures (CNNs, Transformers) and datasets, we demonstrate that (1) incorporating ESD into the training improves model calibration in various batch size settings without the need for internal hyperparameter tuning, (2) ESD yields the best-calibrated results compared with previous approaches, and (3) ESD drastically improves the computational costs required for calibration during training due to the absence of internal hyperparameter. The code is publicly accessible at https://github.com/hee-suk-yoon/ESD.

## 1 Introduction

The calibration of a neural network measures the extent to which its predictions align with the true probability distribution. Possessing this property becomes especially important in real-world applications, such as identification (Kim & Yoo, 2017; Yoon et al., 2022), autonomous driving (Bojarski et al., 2016; Ko et al., 2017), and medical diagnosis (Kocbek et al., 2020; Pham et al., 2022), where uncertainty-based decisions of the neural network are crucial to guarantee the safety of the users. However, despite the success of modern neural networks in accurate classification, they are shown to be poorly calibrated due to the tendency of the network to make predictions with high confidence regardless of the input. (i.e., over-confident predictions) (Guo et al., 2017).

Traditionally, post-processing methods have been used, such as temperature scaling and vector scaling (Guo et al., 2017), to calibrate the model using the validation set after the training by adjusting the logits before the final softmax layer. Various trainable calibration objectives have been proposed recently, such as MMCE (Kumar et al., 2018) and SB-ECE (Karandikar et al., 2021), which are added to the loss function as a regularizer to jointly optimize accuracy and calibration during training. A key advantage of calibration during training is that it is possible to cascade post-processing calibration methods after training to achieve even better-calibrated models. Unfortunately, these existing approaches introduce additional hyperparameters in their proposed calibration objectives, and the performance of the calibration objectives is highly sensitive to these design choices. Therefore these hyperparameters need to be tuned carefully on a per model per dataset basis, which greatly reduces their viability for training on large models and datasets.

---

[*]Equal contribution
[†]Corresponding Author

To this end, we propose Expected Squared Difference (ESD), a trainable calibration objective loss that is hyperparameter-free. ESD is inspired by the KS-Error (Gupta et al., 2021), and it views the calibration error from the perspective of the difference between the two expectations. In detail, our contributions can be summarized as follows:

- We propose ESD as a trainable calibration objective loss that can be jointly optimized with the negative log-likelihood loss (NLL) during training. ESD is a binning-free calibration objective loss, and no additional hyperparameters are required. We also provide an unbiased and consistent estimator of the Expected Squared Difference, and show that it can be utilized in small batch train settings.

- With extensive experiments, we demonstrate that across various architectures (CNNs & Transformers) and datasets (in vision & NLP domains), ESD provides the best calibration results when compared to previous approaches. The calibrations of these models are further improved by post-processing methods.

- We show that due to the absence of an internal hyperparameter in ESD that needs to be tuned, it offers a drastic improvement compared to previous calibration objective losses with regard to the total computational cost for training. The discrepancy in computational cost between ESD and tuning-required calibration objective losses becomes larger as the model complexity and dataset size increases.

## 2 RELATED WORK

Calibration of neural networks has gained much attention following the observation from Guo et al. (2017) that modern neural networks are poorly calibrated. One way to achieve better calibration is to design better neural network architectures tailored for uncertainty estimation, e.g., Bayesian Neural Networks (Blundell et al., 2015; Gal & Ghahramani, 2016) and Deep Ensembles (Lakshminarayanan et al., 2017). Besides model design, **post-processing** calibration strategies have been widely used to calibrate a trained machine learning model using a hold-out validation dataset. Examples include temperature scaling (Guo et al., 2017), which scales the logit output of a classifier with a temperature parameter; Platt scaling (Platt, 1999), which fits a logistic regression model on top of the logits; and Conformal prediction (Vovk et al., 2005; Lei et al., 2018) which uses validation set to estimate the quantiles of a given scoring function. Other post-processing techniques include histogram binning (Zadrozny & Elkan, 2001), isotonic regression (Zadrozny & Elkan, 2002), and Bayesian binning into quantiles (Pakdaman Naeini et al., 2015).

Our work focuses on **trainable calibration methods**, which train neural networks using a hybrid objective, combining a primary training loss with an auxiliary calibration objective loss. In this regard, one popular objective is Maximum Mean Calibration Error (MMCE) (Kumar et al., 2018), which is a kernel embedding-based measure of calibration that is differentiable and, therefore, suitable as a calibration loss. Moreover, Karandikar et al. (2021) proposes a trainable calibration objective loss, SB-ECE, and S-AvUC, which softens previously defined calibration measures.

## 3 PROBLEM SETUP

### 3.1 CALIBRATION ERROR AND METRIC

Let us first consider an arbitrary neural network as $f_\theta : \mathcal{D} \to [0, 1]^C$ with network parameters $\theta$, where $\mathcal{D}$ is the input domain and $C$ is the number of classes in the multiclass classification task. Furthermore, we assume that the training data, $(\boldsymbol{x_i}, \boldsymbol{y_i})_{i=1}^n$ are sampled *i.i.d.* from the joint distribution $\mathbb{P}(\boldsymbol{X}, \boldsymbol{Y})$ (here we use one-hot vector for $\boldsymbol{y}$). Here, $\boldsymbol{Y} = (Y_1, ..., Y_C)$, and $\boldsymbol{y} = (y_1, ..., y_C)$ are samples from this distribution. We can further define a multivariate random variable $\boldsymbol{Z} = f_\theta(\boldsymbol{X})$ as the distribution of the outputs of the neural network. Similarly, $\boldsymbol{Z} = (Z_1, ..., Z_C)$, and $\boldsymbol{z} = (z_1, ..., z_C)$ are samples from this distribution. We use $(z_{K,i}, y_{K,i})$ to denote the output confidence and the one-hot vector element associated with the $K$-th class of the $i$-th training sample. Using this formulation, a neural network is said to be perfectly calibrated for class $K$, if and only if

$$\mathbb{P}(Y_K = 1 | Z_K = z_K) = z_K. \tag{1}$$

Intuitively, Eq. (1) requires the model accuracy for class $K$ to be $z_K$ on average for the inputs where the neural network produces prediction of class $K$ with confidence $z_K$. In many cases, the research of calibration mainly focuses on the calibration of the max output class that the model predicts. Thus, calibration error is normally reported with respect to the predicted class (i.e., max output class) only. As such, $k$ will denote the class with maximum output probability. Furthermore, we also write $I(\cdot)$ as the indicator function which returns one if the Boolean expression is true and zero otherwise.

With these notations, one common measurement of calibration error is the difference between confidence and accuracy which is mathematically represented as (Guo et al., 2017)

$$\mathbb{E}_{Z_k}\left[|\mathbb{P}(Y_k = 1|Z_k) - Z_k|\right]. \tag{2}$$

To estimate this, the Expected Calibration Error (ECE) (Naeini et al., 2015) uses $B$ number of bins with disjoint intervals $B_j = (\frac{j}{B}, \frac{j+1}{B}]$, $j = 0, 1, ..., B-1$ to compute the calibration error as follows:

$$\text{ECE} = \frac{1}{|\mathcal{D}|} \sum_{j=0}^{B-1} \sum_{i=1}^{N} \left| I\left(\frac{j}{B} < z_{k,i} \leq \frac{j+1}{B}, y_{k,i} = 1\right) - z_{k,i}\right|. \tag{3}$$

## 3.2 Calibration During Training

Post-processing method and calibration during training are the two primary approaches for calibrating a neural network. Our focus in this paper is on the latter, where our goal is to train a calibrated yet accurate classifier directly. Note that calibration and predictive accuracy are independent properties of a classifier. In other words, being calibrated does not imply that the classifier has good accuracy and vice versa. Thus, training a classifier to have both high accuracy and good calibration requires jointly optimizing a calibration objective loss alongside the negative log-likelihood (NLL) with a scaling parameter $\lambda$ for the secondary objective:

$$\min_{\theta} \text{NLL}(\mathcal{D}, \theta) + \lambda \cdot \text{CalibrationObjective}(\mathcal{D}, \theta). \tag{4}$$

### 3.2.1 Existing Trainable Calibration Objectives need Tuning

Kumar et al. (2018) and Karandikar et al. (2021) suggested that the disjoint bins in ECE can introduce discontinuities which are problematic when using it as a calibration loss in training. Therefore, additional parameters were introduced for the purpose of calibration during training. For example, Kumar et al. (2018) proposed Maximum Mean Calibration Error (MMCE), where it utilizes a Laplacian Kernel instead of the disjoint bins in ECE. Furthermore, Karandikar et al. (2021) proposed soft-binned ECE (SB-ECE) and soft AvUC, which are softened versions of the ECE metric and the AvUC loss (Krishnan & Tickoo, 2020), respectively. All these approaches address the issue of discontinuity which makes the training objective differentiable. However they all introduce additional design choices - MMCE requires a careful selection of the kernel width $\phi$, SB-ECE needs to choose the number of bins $M$ and a softening parameter $T$, and S-AvUC requires a user-specified entropy threshold $\kappa$ in addition to the softening parameter $T$. Searching for the optimal hyperparameters can be computationally expensive especially as the size of models and dataset become larger.

### 3.2.2 Calibration During Training Suffers from Over-fitting Problem

NLL loss is known to implicitly train for calibration since it is a proper scoring rule, so models trained with NLL can overfit in terms of calibration error (Mukhoti et al., 2020; Karandikar et al., 2021). Figure 1 provides an example of an accuracy curve and its corresponding ECE curve for a model trained with NLL loss. We see that the model is overfitting in terms of both accuracy and ECE, which causes the gap between train and test ECE to become larger during training. As such, adding a calibration objective to the total loss will not be able to improve model calibration as it mainly helps improve calibration of the model with respect to the training data only.

Kumar et al. (2018) tackle this issue by introducing a weighted version of their calibration objective loss. They consider using larger weights to incorrect prediction samples after observing that the fraction of incorrect to the correct samples on the training data is smaller than that of the validation and test data. Instead of changing the calibration objective function itself, Karandikar et al. (2021) introduced a new training scheme called *interleaved training*, where they split the training data

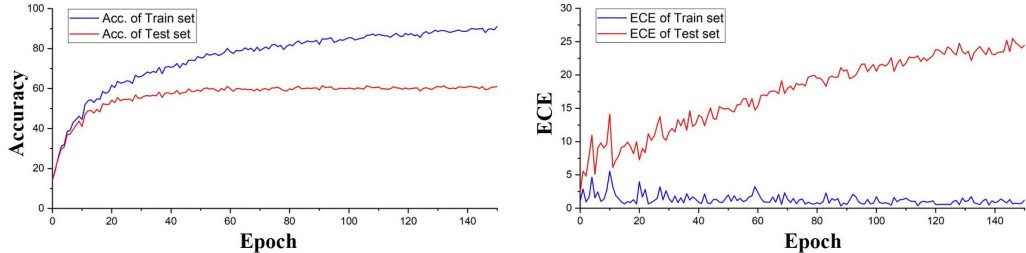

Figure 1: Accuracy (%) curve (left) and its corresponding ECE (%) curve (right) during training with negative log-likelihood (NLL) loss. It could be seen that since NLL implicitly trains for calibration error, the ECE of the train set approaches zero while the ECE of the test set increases during training.

further and dedicate a small portion of it to optimize the calibration objective loss. We will follow this strategy and introduce the technical details in the later sections.

## 4 ESD: EXPECTED SQUARED DIFFERENCE

We propose Expected Squared Difference (ESD) as a tuning-free (i.e., hyperparameter-free) calibration objective. Our approach is inspired by the viewpoint of calibration error as a measure of distance between two distributions to obtain a binning-free calibration metric (Gupta et al., 2021). By using this approach, there is no need to employ kernels or softening operations to handle the bins within calibration metrics, such as ECE, in order to make them suitable for training.

In particular, we consider calibration error as the difference between the two expectations. We first start with the definition of perfect calibration in Eq. (1):

$$
\begin{aligned}
& \mathbb{P}(Y_k = 1 | Z_k = z_k) = z_k && \forall z_k \in [0,1] \\
\Leftrightarrow\ & \mathbb{P}(Y_k = 1, Z_k = z_k) = z_k \mathbb{P}(Z_k = z_k) && \forall z_k \in [0,1] \qquad \text{(by Bayes rule)}.
\end{aligned}
\tag{5}
$$

Now considering the accumulation of the terms on both sides for arbitrary confidence level $\alpha \in [0,1]$, the perfect calibration for an arbitrary class $k$ can now be written as:

$$
\begin{aligned}
& \int_0^\alpha \mathbb{P}(Y_k = 1, Z_k = z_k)dz_k = \int_0^\alpha z_k \mathbb{P}(Z_k = z_k)dz_k \\
\Leftrightarrow\ & \mathbb{E}_{Z_k, Y_k}[I(Z_k \le \alpha, Y_k = 1)] = \mathbb{E}_{Z_k, Y_k}[Z_k I(Z_k \le \alpha)], \quad \forall \alpha \in [0,1].
\end{aligned}
\tag{6}
$$

This allows us to write the difference between the two expectations as

$$
\begin{aligned}
d_k(\alpha) &= \left| \int_0^\alpha \mathbb{P}(Y_k = 1, Z_k = z_k) - z_k \mathbb{P}(Z_k = z_k)dz_k \right| \\
&= |\mathbb{E}_{Z_k, Y_k}[I(Z_k \le \alpha)(I(Y_k = 1) - Z_k)]|,
\end{aligned}
\tag{7}
$$

and $d_k(\alpha) = 0, \forall \alpha \in [0,1]$ if and only if the model is perfectly calibrated for class $k$. Since this has to hold $\forall \alpha \in [0,1]$, we propose ESD as the expected squared difference between the two expectations:

$$
\mathbb{E}_{Z_k'}[d_k(Z_k')^2] = \mathbb{E}_{Z_k'}[\mathbb{E}_{Z_k, Y_k}^2[I(Z_k \le Z_k')(I(Y_k = 1) - Z_k)]].
\tag{8}
$$

Due to the close relationship between $d_k(\alpha)$ and calibration of a neural network, the difference between an uncalibrated and a calibrated neural network can be clearly observed using $\mathbb{E}_{Z_k'}[d_k(Z_k')^2]$ as visually shown in Appendix A. Since ESD = 0 iff. the model is perfectly calibrated as shown in the following theorem, this metric is a good measure of calibration.

**Theorem 1.** $\mathbb{E}_{Z_k'}[\mathbb{E}_{Z_k, Y_k}^2[I(Z_k \le Z_k')(I(Y_k = 1) - Z_k)]] = 0$ *iff. the model is perfectly calibrated.*

*Proof.* Since $d_k(Z_k')^2 = \mathbb{E}_{Z_k, Y_k}^2[I(Z_k \le Z_k')(I(Y_k = 1) - Z_k)]$ is a non-negative random variable induced by $Z_k'$, $\mathbb{E}_{Z_k'}[d_k(Z_k')^2] = 0$ iff. $\mathbb{P}(d_k(Z_k') = 0) = 1$. Furthermore,

$$
\mathbb{P}(d_k(Z_k') = 0) = 1 \quad \Longleftrightarrow \quad d_k(\alpha) = 0 \quad \forall \alpha \in \mathcal{I} \text{ where } \mathcal{I} \text{ is the support set of } Z_k'.
$$

Thus,

$$\mathbb{E}_{Z'_k}[d_k(Z'_k)^2] = 0 \text{ iff. } d_k(\alpha) = 0 \quad \forall \alpha \in \mathcal{I}.$$

From lemma 1.1 below, we have $d_k(\alpha) = 0 \ \forall \alpha \in [0,1]$ iff. $d_k(\alpha) = 0 \ \forall \alpha$ s.t. $\mathbb{P}(Z'_k = \alpha) \neq 0$. Consequently, $\mathbb{E}_{Z'_k}[\mathbb{E}^2_{Z_k,Y_k}[I(Z_k \leq Z'_k)(I(Y_k = 1) - Z_k)]] = 0$ iff. the model is perfectly calibrated. $\qquad \square$

**Lemma 1.1** *Let $\mathcal{I}$ be the support set of random variable $Z_k$, then $d_k(\alpha) = 0 \ \forall \alpha \in \mathcal{I}$ iff. $d_k(\alpha) = 0$ $\forall \alpha \in [0,1]$.*

*Proof.* The backward direction result is straight-forward, so we only prove for the forward direction:
If $d_k(\alpha) = 0 \ \forall \alpha \in \mathcal{I}$, then $d_k(\alpha) = 0 \ \forall \alpha \in [0,1]$.
For arbitrary $\alpha' \in \mathcal{I}^c$, let $\alpha = \arg\min_{a \in \mathcal{I}} |a - \alpha'|$ and $\alpha < \alpha'$ . We then have,

$$
\begin{aligned}
d_k(\alpha') &= |\int_0^{\alpha'} \mathbb{P}(Y_k = 1, Z_k = z_k) - z_k \mathbb{P}(Z_k = z_k) dz_k| \\
&= |\int_0^{\alpha'} \mathbb{P}(Y_k = 1|Z_k = z_k)\mathbb{P}(Z_k = z_k) - z_k \mathbb{P}(Z_k = z_k) dz_k| \\
&= |\int_0^{\alpha} \mathbb{P}(Y_k = 1|Z_k = z_k)\mathbb{P}(Z_k = z_k) - z_k \mathbb{P}(Z_k = z_k) dz_k \\
&\quad + \int_{\alpha}^{\alpha'} \mathbb{P}(Y_k = 1|Z_k = z_k)\mathbb{P}(Z_k = z_k) - z_k \mathbb{P}(Z_k = z_k) dz_k| \\
&= |\int_0^{\alpha} \mathbb{P}(Y_k = 1|Z_k = z_k)\mathbb{P}(Z_k = z_k) - z_k \mathbb{P}(Z_k = z_k) dz_k| \\
&= 0.
\end{aligned}
$$

$\qquad \square$

### 4.1 AN ESTIMATOR FOR ESD

In this section, we use $(z_{k,i}, y_{k,i})$ to denote the output confidence and the one-hot vector element associated with the $k$-th class of the $i$-th training sample respectively. As the expectations in the true Expected Squared Difference (Eq. (8)) are intractable, we propose a Monte Carlo estimator for it which is unbiased. A common approach is to use a naive Monte Carlo sampling with respect to both the inner and outer expectations to give the following:

$$\mathbb{E}_{Z'_k}[\mathbb{E}^2_{Z_k,Y_k}[I(Z_k \leq Z'_k)(I(Y_k = 1) - Z_k)]] \approx \frac{1}{N}\sum_{i=1}^{N} \bar{g}_i^{\;2}, \tag{9}$$

$$\text{where } \bar{g}_i = \frac{1}{N-1}\sum_{\substack{j=1 \\ j \neq i}}^{N} g_{ij} \text{ and } g_{ij} = I(z_{k,j} \leq z_{k,i})[I(y_j = k) - z_{k,j}].$$

However, Eq. (9) results in a biased estimator that is an upper bound of the true Expected Squared Difference. To account for the bias, we propose the unbiased and consistent estimator (proof in Appendix B) of the true Expected Squared Difference[1],

$$ESD = \frac{1}{N}\sum_{i=1}^{N}\left[\bar{g}_i^{\;2} - \frac{S_{g_i}^2}{N-1}\right] \quad \text{where } S_{g_i}^2 = \frac{1}{N-2}\sum_{\substack{j=1 \\ j \neq i}}^{N}(g_{ij} - \bar{g}_i)^2. \tag{10}$$

---

[1]To avoid confusion, ESD from this point onward will refer to the estimator instead of its expectation form.

## 4.2 INTERLEAVED TRAINING

Negative log-likelihood (NLL) has be shown to greatly overfit to ECE of the data it is trained on. Thus, training for calibration using the same data for the NLL has limited effect on reducing the calibration error of the model. Karandikar et al. (2021) proposed *interleaved training* where they split the train set into two subsets - one is used to optimize the NLL and the other is used to optimize the calibration objective. Following this framework, let $\mathcal{D}_{train}$ denote the entire train set. We separate the train set into two subsets - $\mathcal{D}'_{train}$ and $\mathcal{D}'_{cal}$. The joint training of ESD with NLL becomes,

$$\min_{\theta} \text{NLL}(\mathcal{D}'_{train}, \theta) + \lambda \cdot \text{ESD}(\mathcal{D}'_{cal}, \theta). \tag{11}$$

With this training scheme, NLL is optimized on $\mathcal{D}'_{train}$ and ESD is optimized on $\mathcal{D}'_{cal}$. This way, Karandikar et al. (2021) showed we can avoid minimizing ECE that is already overfit to the train set.

## 5 EXPERIMENTAL SETTING

### 5.1 DATASETS AND MODELS

**Image Classification**    For image classification tasks we use the following datasets:

- MNIST (Deng, 2012): 54,000/6,000/10,000 images for train, validation, and test split was used. We resized the images to (32x32) before inputting to the network.
- CIFAR10 & CIFAR100 (Krizhevsky et al., a;b): 45,000/5,000/10,000 images for train, validation, and test split was used. Used random cropping of 32 with padding of 4. Normalized each RGB channel with mean of 0.5 and standard deviation of 0.5.
- ImageNet100 (Deng et al., 2009): A subset dataset from ImageNet Large Scale Visual Recognition Challenge 2012 with 100 classes. Since the labels of test sets are unavailable, we use the official validation set as the test set, and we dedicate 10% of the training data as the validation set for the experiments (i.e., 117,000/13,000/5,000 split for train/val/test set).

**Natural Language Inference (NLI)**    NLI is a task in Natural Language Processing where it involves classifying the inference relation (entailment, contradiction, or neutral) between two texts (MacCartney & Manning, 2008). For NLI tasks we use the following datasets:

- SNLI (Bowman et al., 2015): SNLI corpus is a collection of human-written English sentence pairs manually labeled for balanced classification with the labels entailment, contradiction, and neutral. The data consists of 550,152/10,000/10,000 sentence pairs for train/val/test set respectively. The max length of the input is set to 158.
- ANLI (Nie et al., 2020): ANLI dataset is a large-scale NLI dataset, collected via an, adversarial human-and-model-in-the-loop procedure. The data consists of 162,865/3,200/3,200 sentence pairs for train/val/test set respectively. The max length of the input is set to 128.

For the Image Classification datasets, we used Convolutional Neural Networks (CNNs). Specifically, we used LeNet5 (Lecun et al., 1998), ResNet50, ResNet34, and ResNet18 (He et al., 2016), for MNIST, CIFAR10, CIFAR100, and ImageNet100, respectively. For the NLI datasets, we finetuned transformer based Pre-trained Language Models (PLMs). Specifically, we used Bert-base (Devlin et al., 2019) and Roberta-base (Liu et al.), for SNLI and ANLI, respectively.

### 5.2 EXPERIMENTAL SETUP

We compare our Expected Squared Difference (ESD) to previously proposed trainable calibration objectives, MMCE and SB-ECE, on the datasets and models previously mentioned. For fair comparison, interleaved training has been used for all three calibration objectives. For MMCE, in which Kumar et al. (2018) proposed an unweighted and weighted version of the objective, we use the former to set the method to account for the overfitting problem mentioned in section 3.2.2 consistent to interleaved training. For the interleaved training settings, we held out 10% of the train set to the calibration set. The regularizer hyperparameter $\lambda$ for weighting the calibration measure with respect to NLL is chosen via fixed grid search [2]. For measuring calibration error, we use ECE with 20 equally sized

---

[2]For $\lambda$, we search for [0.2, 0.4, 0.6, 0.8, 1.0, 2.0, 3.0, ... 10.0] (Appendix E).

| Dataset (Model) | Loss Fn. | Acc. | ECE | ECE *after* TS | Acc. *after* VS | ECE *after* VS |
|---|---|---|---|---|---|---|
| MNIST (LeNet5) | NLL (baseline) | 98.8±0.034 | 0.91±0.080 | 0.31±0.044 | 98.7±0.058 | 0.43±0.123 |
| | +MMCE | 98.4±0.320 | 0.36±0.029 | 0.33±0.068 | 98.4±0.250 | 0.32±0.055 |
| | +SB-ECE | 97.9±0.177 | 0.41±0.067 | 0.45±0.103 | 97.9±0.073 | 0.39±0.088 |
| | **+ESD (ours)** | 98.6±0.204 | **0.30**±0.035 | **0.29**±0.030 | 98.6±0.167 | **0.28**±0.071 |
| CIFAR10 (Resnet50) | NLL (baseline) | 92.9±0.159 | 5.49±0.105 | 1.89±0.105 | 92.3±0.625 | 2.96±1.596 |
| | +MMCE | 91.5±0.340 | 4.92±0.292 | 1.92±0.274 | 91.3±0.429 | 2.52±0.586 |
| | +SB-ECE | 91.6±0.288 | 4.86±0.319 | 1.63±0.300 | 91.5±0.343 | 1.88±0.393 |
| | **+ESD (ours)** | 92.1±0.141 | **3.08**±0.692 | **1.60**±0.089 | 92.1±0.314 | **1.61**±0.287 |
| CIFAR100 (Resnet34) | NLL (baseline) | 68.4±0.491 | 23.8±0.403 | 5.74±0.306 | 67.3±0.551 | 9.78±0.640 |
| | +MMCE | 66.5±0.644 | 13.9±0.545 | 4.91±0.457 | 66.3±0.773 | 4.59±1.575 |
| | +SB-ECE | 67.3±0.367 | 14.7±1.370 | 4.94±0.240 | 66.4±0.369 | 4.57±1.253 |
| | **+ESD (ours)** | 67.4±0.356 | **13.6**±0.950 | **4.85**±0.390 | 67.1±0.496 | **4.28**±1.396 |
| ImageNet100 (Resnet18) | NLL (baseline) | 75.8±0.397 | 10.3±0.686 | 2.51±0.378 | 75.9±0.595 | 2.86±0.459 |
| | +MMCE | 74.3±0.248 | 3.83±0.644 | 1.94±0.262 | 74.5±0.444 | 2.29±0.227 |
| | +SB-ECE | 74.4±0.596 | 4.28±0.318 | 2.06±0.260 | 74.7±0.472 | 2.13±0.198 |
| | **+ESD (ours)** | 74.6±0.320 | **1.80**±0.262 | **1.72**±0.350 | 74.8±0.436 | **1.87**±0.266 |
| SNLI (Bert-base) | NLL (baseline) | 90.2±0.434 | 4.20±0.420 | 1.04±0.112 | 90.1±0.397 | 0.96±0.099 |
| | +MMCE | 89.4±0.491 | 1.11±0.181 | 1.01±0.178 | 89.5±0.588 | 1.08±0.170 |
| | +SB-ECE | 89.1±0.668 | 1.82±0.301 | 0.99±0.180 | 89.2±0.765 | 0.90±0.109 |
| | **+ESD (ours)** | 89.3±0.536 | **0.98**±0.165 | **0.73**±0.192 | 89.4±0.583 | **0.61**±0.126 |
| ANLI (Roberta-base) | NLL (baseline) | 49.4±0.323 | 35.9±0.505 | 4.16±0.445 | 48.4±0.492 | 5.10±0.657 |
| | +MMCE | 49.0±0.471 | 31.2±0.850 | 3.71±0.175 | 47.7±0.429 | 4.79±0.693 |
| | +SB-ECE | 48.5±0.481 | 33.9±1.378 | 3.98±0.733 | 47.3±0.281 | 5.10±0.119 |
| | **+ESD (ours)** | 48.0±0.451 | **28.8**±0.543 | **3.49**±0.373 | 47.1±0.429 | **4.42**±1.010 |

Table 1: Average accuracy (%) and ECE (%) (with std. across 5 trials) for baseline, MMCE, SB-ECE, ESD after training and after post-processing with temperature scaling (TS) or vector scaling (VS).

bins. For the image classification tasks we use AdamW (Loshchilov & Hutter, 2019) optimizer with $10^{-3}$ learning rate and $10^{-2}$ weight decay for 250 epochs, except for ImageNet100, in which case we used $10^{-4}$ weight decay for 90 epochs. For the NLI tasks, we use AdamW optimizer with $10^{-5}$ learning rate and $10^{-2}$ weight decay for 15 epochs. For both tasks, we use a batch size of 512.

The internal hyperparameters within MMCE ($\phi$) and SB-ECE ($M$, $T$) were sequentially optimized [3] following the search for optimal $\lambda$. Following the original experimental setting of SB-ECE by Karandikar et al. (2021), we fix the hyperparameter $M$ to 15. Similar to the model selection criterion utilized by Karandikar et al. (2021), we look at the accuracy and the ECE of all possible hyperparameter configurations in the grid. We choose the lowest ECE while giving up less than 1.5% accuracy relative to baseline accuracy on the validation set. All experiments were done using NVIDIA Quadro RTX 8000 and NVIDIA RTX A6000.

## 6 EXPERIMENTAL RESULT

In Table 1, we report the accuracy and ECE for the models before and after post-processing (i.e., temperature scaling and vector scaling) of various datasets and models. Compared to the baseline or other trainable calibration objective loss (MMCE and SB-ECE), jointly training with ESD as the secondary loss consistently results in a better-calibrated network for all the datasets and models with around 1% degradation of accuracy. Moreover, we observe that applying post-processing (i.e., temperature scaling (TS) and vector scaling (VS)) after training with a calibration objective loss generally results in better-calibrated models over baseline post-processing, in which case ESD still outperforms other methods. Comparing the calibration outcomes in temperature scaling with vector scaling for models trained with ESD, vector scaling worked comparable if not better as a post-processing method, except for ANLI, with minimal impact on the accuracy for all datasets. Additionally, Ovadia et al. (2019) demonstrated that postprocessing methods, such as temperature

---

[3]For $\phi$, we search [0.2, 0.4, 0.6, 0.8]. For $T$, we search [0.0001, 0.001, 0.01, 0.1].

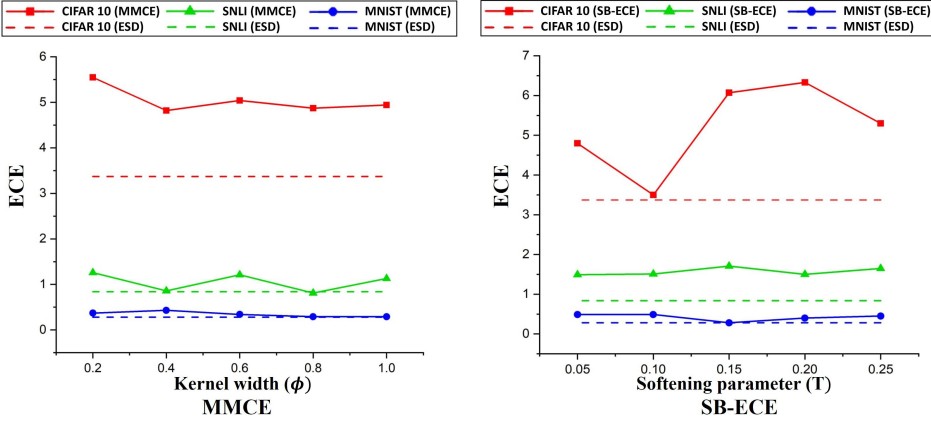

Figure 2: ECE performance curve of MMCE (left) and SB-ECE (right) with respect to their varying internal hyperparameters on MNIST, CIFAR10, SNLI datasets.

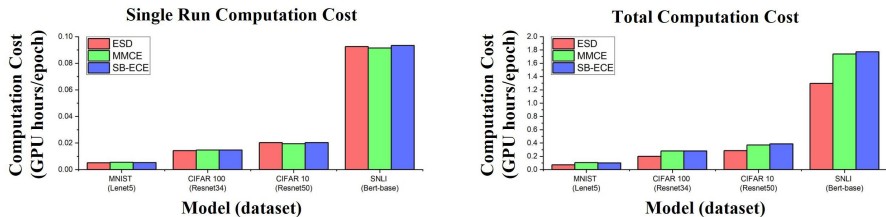

Figure 3: Computational cost of single-run training (left) and total cost considering hyperparameter tuning (right). The x-axis in both cases are in the order of increasing model complexity.

scaling, are not effective in achieving good calibration under distribution shifts. Building on this research, Karandikar et al. (2021) have provided evidence that incorporating calibration during model training can enhance a model's ability to maintain calibration under distribution shifts. This finding has been replicated in Appendix F.

We show in Figure 2 that across different models and datasets the calibration performance of MMCE and SB-ECE is sensitive to the internal hyperparameter. This shows the importance of hyperparameter tuning in these methods. On the other hand, ESD does not only outperforms MMCE and SB-ECE in terms of ECE but does not have an internal hyperparameter to tune for.

## 7 Ablation Study

### 7.1 Computational Cost of Hyperparameter Seach

We investigate the computational cost required to train a model with ESD compared to MMCE and SB-ECE. As depicted in Figure 3, the computational cost required to train a model for a single run remains nearly identical across different models and datasets. However, considering the need for tuning additional hyperparameters within MMCE and SB-ECE, the discrepancy in total computational cost between ESD and tuning-required calibration objective losses becomes more prominent as the model complexity and dataset size increases.

### 7.2 Batch Size Experiments

Table 2 shows that ESD is robust to varying batch sizes which could be due to the fact that it is an unbiased and consistent estimator. Regardless of the batch size, being unbiased guarantees that the average of the gradient vectors of ESD equals the gradient of the true estimate (Appendix D).

| Dataset (Model) | Loss Fn. | Acc. | ECE | ECE *after* TS | Acc. *after* VS | ECE *after* VS |
|---|---|---|---|---|---|---|
| CIFAR10 (Resnet50) | NLL (b512) | 92.9±0.159 | 5.49±0.105 | 1.89±0.105 | 92.3±0.625 | 2.96±1.596 |
| | +ESD (b512) | 92.1±0.141 | 3.08±0.692 | 1.60±0.089 | 92.1±0.314 | 1.61±0.287 |
| | +ESD (b256) | 91.5±0.137 | 3.15±0.587 | 1.61±0.073 | 91.9±0.573 | 1.60±0.082 |
| | +ESD (b128) | 91.5±0.519 | 2.93±0.539 | 1.70±0.600 | 91.7±0.246 | 2.20±0.584 |
| | +ESD (b64) | 91.3±0.753 | 3.91±0.847 | 1.87±0.132 | 91.5±0.257 | 2.65±1.48 |
| SNLI (Bert-base) | NLL (b512) | 90.2±0.434 | 4.20±0.420 | 1.04±0.112 | 90.1±0.397 | 0.96±0.099 |
| | +ESD (b512) | 89.3±0.536 | 0.98±0.165 | 0.73±0.192 | 89.4±0.583 | 0.61±0.126 |
| | +ESD (b256) | 89.9±0.314 | 1.11±0.236 | 0.76±0.241 | 88.9±0.146 | 0.62±0.147 |
| | +ESD (b128) | 89.8±0.435 | 0.99±0.244 | 0.68±0.190 | 89.6±0.490 | 0.58±0.187 |
| | +ESD (b64) | 89.1±0.660 | 1.06±0.285 | 0.83±0.260 | 88.8±0.672 | 0.81±0.153 |

Table 2: Average accuracy and ECE (with std. across 5 trials) for ESD after training with batch sizes 64, 128, 256, and 512.

## 7.3 ARE INDICATOR FUNCTIONS TRAINABLE?

Recent papers, Kumar et al. (2018) and Karandikar et al. (2021), have suggested that ECE is not suitable for training as a result of its high discontinuity due to binning, which can be seen as a form of an indicator function. However, the results from our method suggest that our measure was still able to train well despite the existence of indicator functions. In addition, previous measures also contain indicator functions in the form of *argmax* function that introduces discontinuities but remains to be trainable. As such, this brings to rise the question of whether calibration measures with indicator functions can be used for training. To investigate this, we ran ECE as an auxiliary calibration loss on different batch sizes and observed its performance on CIFAR 100 (Table 3).

We found that ECE, contrary to previous belief, is trainable under large batch sizes and not trainable under small batch sizes while ours maintains good performance regardless of batch size. The poor performance of ECE under small batch size setting could potentially be attributed to the high bias present in such cases. From this, it seems to suggest that indicator functions do not seem to inhibit the training for calibration.

| Loss Fn. | Acc. | ECE |
|---|---|---|
| NLL (b512) | 68.6±0.034 | 22.4±0.105 |
| +ECE (b512) | 68.2±0.025 | 14.6±0.252 |
| +ECE (b256) | 67.9±0.057 | 18.1±0.552 |
| +ECE (b128) | 68.1±0.084 | 21.0±0.552 |

Table 3: Average accuracy and ECE (with std. across 5 trials) for CIFAR100 after training with ECE on batch sizes 128, 256, and 512.

## 8 CONCLUSIONS

Motivated by the need for a tuning-free trainable calibration objective, we proposed Expected Squared Difference (ESD), which does not contain any internal hyperparameters. With extensive comparison with existing methods for calibration during training across various architectures (CNNs & Transformers) and datasets (in vision & NLP domains), we demonstrate that training with ESD provides the best-calibrated models compared to other methods with minor degradation in accuracy over baseline. Furthermore, the calibration of these models is further improved after post-processing. In addition, we demonstrate that ESD can be utilized in small batch settings while maintaining performance. More importantly, in contrast to previously proposed trainable calibration objectives, ESD does not contain any internal hyperparameters, which significantly reduces the total computational cost for training. This reduction in cost is more prominent as the complexity of the model and dataset increases, making ESD a more viable calibration objective option.

ACKNOWLEDGEMENT

This work was supported by Institute of Information & communications Technology Planning & Evaluation (IITP) grant funded by the Korea government(MSIT) (No.2022-0-00184, Development and Study of AI Technologies to Inexpensively Conform to Evolving Policy on Ethics), and Institute for Information & communications Technology Promotion(IITP) grant funded by the Korea government(MSIT) (No. 2021-0-01381, Development of Causal AI through Video Understanding and Reinforcement Learning, and Its Applications to Real Environments).

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

## A  VISUAL INTUITION OF EXPECTED SQUARED DIFFERENCE (ESD)

From Eq. (7),

$$d_k(\alpha) = \left| \int_0^\alpha \mathbb{P}(Y_k = 1, Z_k = z_k) - z_k \mathbb{P}(Z_k = z_k) dz_k \right|$$
$$= |\mathbb{E}_{Z_k, Y_k}[I(Z_k \le \alpha)(I(Y_k = 1) - Z_k)]|.$$

This can be viewed as the difference between two quantities:

$$\textbf{Cumulative Accuracy} = \int_0^\alpha \mathbb{P}(Y_k = 1, Z_k = z_k) dz_k$$
$$= \mathbb{E}_{Z_k, Y_k}[I(Z_k \le \alpha, Y_k = 1)]$$
$$\approx \frac{1}{N} \sum_{i=1}^N I(z_{k,i} \le \alpha, Y_{k,i} = 1).$$
$$\textbf{Cumulative Confidence} = \int_0^\alpha z_k \mathbb{P}(Z_k = z_k) dz_k$$
$$= \mathbb{E}_{Z_k, Y_k}[Z_k I(Z_k \le \alpha)]$$
$$\approx \frac{1}{N} \sum_{i=1}^N z_{k,i} I(z_{k,i} \le \alpha).$$

Due to the close relationship between $d_k(\alpha)$ and the calibration of a neural network, the average squared difference ($\mathbb{E}_{Z_k'}[d_k(Z_k')^2]$) between the cumulative accuracy and confidence closely corresponds with the calibration of a network (Figure 4). That is, the average squared difference between the cumulative accuracy and confidence is larger for an uncalibrated network compared to a calibrated network. Jointly training with our proposed Expected Squared Difference (ESD) as an auxiliary loss tries to minimize this squared difference between the two curves during training on average, thus achieving a better calibrated model.

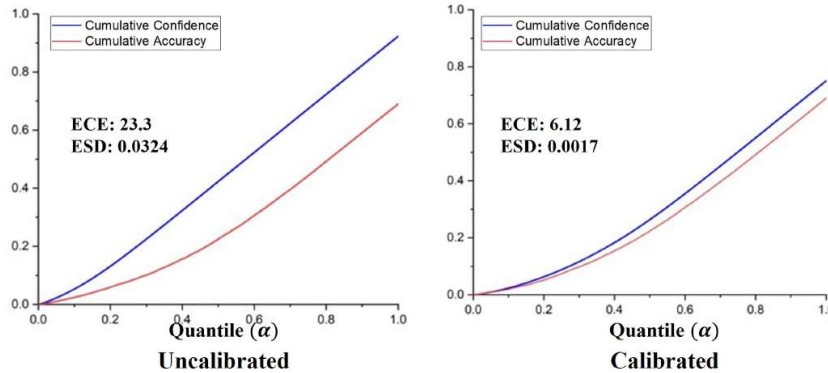

Figure 4: Visual intuition plot showing the cumulative confidence and cumulative accuracy with varying quantile scores of prediction confidence ($\alpha$) for an uncalibrated (left) and calibrated (right) network. The uncalibrated network was obtained by training Resnet34 on CIFAR100 with NLL, and the calibrated network was acquired by temperature scaling on the aforementioned trained network.

## B  PROOF THAT ESD IS AN UNBIASED AND CONSISTENT ESTIMATOR

In Theorem 2, we prove that ESD is an unbiased estimator, that is taking the expectation of ESD is equal to the true Expected Squared Difference.

**Theorem 2** *ESD is an unbiased estimator, i.e.* $\mathbb{E}_{Z_k, Y_k}[ESD] = \mathbb{E}_{Z_k'}[d_k^2(Z_k')]]$ *where* $d_k(Z_k') = |\mathbb{E}_{Z_k, Y_k}[I(Z_k \le Z_k')(I(Y_k = 1) - Z_k)]|.$

*Proof.* Let $\boldsymbol{Z_k}=(Z_{k,1},...,Z_{k,n})$, $\boldsymbol{Y_k} = (Y_{k,1},...,Y_{k,n})$ and $G_i = \bar{g_i}^2 - \frac{S_{g_i}^2}{N-1}$. We have that

$$
\begin{aligned}
\mathbb{E}_{\boldsymbol{Z_k},\boldsymbol{Y_k}}[ESD] &= \frac{1}{N}\sum_{i=1}^{N}\mathbb{E}_{\boldsymbol{Z_k},\boldsymbol{Y_k}}[G_i] \quad \text{(by linearity of expectation)} \\
&= \frac{1}{N}\sum_{i=1}^{N}\mathbb{E}_{Z_{k,i},Y_{k,i}}[\mu_i^2] \quad \text{(by lemma 2.1)} \\
&= \frac{1}{N}\sum_{i=1}^{N}\mathbb{E}_{Z_k'}[d_k^2(Z_k')] \quad \text{(since } \mathbb{E}_{Z_{k,i},Y_{k,i}}[\mu_i^2] = \mathbb{E}_{Z_k'}[d_k^2(Z_k')]\text{)} \\
&= \mathbb{E}_{Z_k'}[d_k^2(Z_k')].
\end{aligned}
$$

$\square$

**Lemma 2.1** $\mathbb{E}_{\boldsymbol{Z_{k,-i}},\boldsymbol{Y_{k,-i}}}[G_i] = \mu_i^2$ *where* $\mu_i = \mathbb{E}_{Z_k,Y_k}[I(Z_k \le Z_{k,i})(I(Y_k = 1) - z_k)]$.

*Proof.* Since samples are i.i.d., for a fixed $i$, it holds that:

$$
\begin{aligned}
\mathbb{E}_{\boldsymbol{Z_{k,-i}},\boldsymbol{Y_{k,-i}}}[\bar{g_i}] &= \mu_i \\
\mathbb{E}_{\boldsymbol{Z_{k,-i}},\boldsymbol{Y_{k,-i}}}\left[\frac{S_{g_i}^2}{N-1}\right] &= \frac{\sigma_i^2}{N-1} \quad \text{(where Var}[g_i] = \sigma_i^2\text{)} \qquad (12)\\
\text{Var}[\bar{g_i}] &= \mathbb{E}_{\boldsymbol{Z_{k,-i}},\boldsymbol{Y_{k,-i}}}[\bar{g_i}^2] - \mu_i^2.
\end{aligned}
$$

Shifting variables around, we get

$$
\begin{aligned}
\mu_i^2 &= \mathbb{E}_{\boldsymbol{Z_{k,-i}},\boldsymbol{Y_{k,-i}}}[\bar{g_i}^2] - \text{Var}[\bar{g_i}] \\
&= \mathbb{E}_{\boldsymbol{Z_{k,-i}},\boldsymbol{Y_{k,-i}}}\left[\bar{g_i}^2 - \frac{S_{g_i}^2}{N-1}\right] \quad \text{(since Var}[\bar{g_i}] = \frac{\sigma_i^2}{N-1}\text{)}\\
&= \mathbb{E}_{\boldsymbol{Z_{k,-i}},\boldsymbol{Y_{k,-i}}}[G_i].
\end{aligned}
$$

$\square$

In Theorem 3, we prove that ESD is a consistent estimator, which means that ESD converges in probability to the true Expected Squared Difference.

**Theorem 3** *ESD is a consistent estimator, i.e. ESD* $\xrightarrow{\mathbb{P}} \mathbb{E}_{Z_k'}[d_k^2(Z_k')]$.

*Proof.* Since ESD is an unbiased estimator, it is sufficient to prove $\lim_{n \to \infty} \text{Var}[ESD] = 0$. For simplicity, we use where $\sum_{i,j=1}^{N} = \sum_{i=1}^{N} \sum_{j=1}^{N}$ and $\mathbb{E}_{\boldsymbol{Z_k}, \boldsymbol{Y_k}}[G_i] = \mathbb{E}_{Z'_k}[d_k^2(Z'_k)] \, \forall i$,

$$\text{Var}[ESD] = \frac{1}{N^2} \sum_{i=1}^{N} \text{Var}[G_i] + \frac{1}{N^2} \sum_{\substack{i,j=1 \\ i \neq j}}^{N} \text{Cov}[G_i, G_j].$$

Since $\mathbb{E}_{\boldsymbol{Z_k}, \boldsymbol{Y_k}}[G_i] = \mathbb{E}_{Z'_k}[d_k^2(Z'_k)] \, \forall i$,

$$= \frac{1}{N^2} \sum_{i=1}^{N} \text{Var}[G_i] + \frac{1}{N^2} \sum_{\substack{i,j=1 \\ i \neq j}}^{N} \left( \mathbb{E}_{\boldsymbol{Z_k}, \boldsymbol{Y_k}}[G_i G_j] - \mathbb{E}_{Z'_k}^2[d_k^2(Z'_k)] \right)$$

$$\leq \frac{1}{N^2} \sum_{i=1}^{N} 2 + \frac{1}{N^2} \sum_{\substack{i,j=1 \\ i \neq j}}^{N} \left( \mathbb{E}_{\boldsymbol{Z_k}, \boldsymbol{Y_k}}[\bar{g_i}^2 \bar{g_j}^2] - \mathbb{E}_{Z'_k}^2[d_k^2(Z'_k)] \right) \quad \text{(by lemma 3.1 and 3.2)}$$

$$+ \frac{1}{N^2} \sum_{\substack{i,j=1 \\ i \neq j}}^{N} \frac{4}{(N-2)^2}$$

$$= \frac{2}{N} + \frac{4N(N-1)}{N^2(N-2)^2} + \frac{N(N-1)}{N^2} \left( \mathbb{E}_{\boldsymbol{Z_k}, \boldsymbol{Y_k}}[\bar{g_1}^2 \bar{g_2}^2] - \mathbb{E}_{Z'_k}^2[d_k^2(Z'_k)] \right). \text{ (by lemma 3.3)}$$

Thus, by using lemma 3.4 and the non-negativeness of variance,

$$0 \leq \lim_{n \to \infty} \text{Var}[ESD] \leq \lim_{n \to \infty} \mathbb{E}_{\boldsymbol{Z_k}, \boldsymbol{Y_k}}[\bar{g_1}^2 \bar{g_2}^2] - \mathbb{E}_{Z'_k}^2[d_k^2(Z'_k)] = 0.$$

Consequently, $\lim_{n \to \infty} \text{Var}[ESD] = 0$.

$\square$

**Lemma 3.1** $Var[G_i] \leq 2 < \infty$.

*Proof.*

$$\text{Var}[G_i] = \mathbb{E}_{\boldsymbol{Z_k}, \boldsymbol{Y_k}}[G_i^2] - \mathbb{E}_{\boldsymbol{Z_k}, \boldsymbol{Y_k}}^2[G_i]$$

$$\leq \mathbb{E}_{\boldsymbol{Z_k}, \boldsymbol{Y_k}}[G_i^2]$$

$$= \mathbb{E}_{\boldsymbol{Z_k}, \boldsymbol{Y_k}} \left[ \bar{g_i}^2 - 2 \frac{S_{g_i}^2}{N-1} + \left( \frac{S_{g_i}^2}{N-1} \right)^2 \right]$$

$$\leq \mathbb{E}_{\boldsymbol{Z_k}, \boldsymbol{Y_k}} \left[ \bar{g_i}^2 + \left( \frac{S_{g_i}^2}{N-1} \right)^2 \right].$$

By lemma 3.2, $\mathbb{E}_{\boldsymbol{Z_k}, \boldsymbol{Y_k}}[g_i^2] \leq 1$ and $\mathbb{E}_{\boldsymbol{Z_k}, \boldsymbol{Y_k}} \left[ \left( \frac{S_{g_i}^2}{N-1} \right)^2 \right] \leq \left( \frac{2}{N-2} \right)^2 \leq 1$. Thus,

$$\mathbb{E}_{\boldsymbol{Z_k}, \boldsymbol{Y_k}} \left[ \bar{g_i}^2 + \left( \frac{S_{g_i}^2}{N-1} \right)^2 \right] \leq 2.$$

$\square$

**Lemma 3.2** $|\frac{S_{g_i}^2}{N-1}| \leq \frac{2}{N-2}$ *and* $|\bar{g_i}| \leq 1$.

*Proof.* By the triangle inequality, $|\bar{g}_i| = |\frac{1}{N-1} \sum\limits_{\substack{m=1 \\ m \neq i}}^{N} g_{im}| \leq \frac{1}{N-1} \sum\limits_{\substack{m=1 \\ m \neq i}}^{N} |g_{im}| \leq 1$, since $\forall m \ |g_{im}| \leq 1$.

Furthermore, we have

$$
\begin{aligned}
\left| \frac{S_{g_i}^2}{N-1} \right| &= \frac{1}{(N-1)^2} \sum_{\substack{m=1 \\ m \neq i}}^{N} g_{im}^2 + \frac{1}{(N-1)^2(N-2)} \Big( \sum_{\substack{m=1 \\ m \neq i}}^{N} g_{im} \Big)^2 \\
&\leq \frac{1}{(N-1)^2} \sum_{\substack{m=1 \\ m \neq i}}^{N} |g_{im}|^2 + \frac{1}{(N-1)^2(N-2)} \Big( \sum_{\substack{m=1 \\ m \neq i}}^{N} |g_{im}| \Big)^2 \\
&\leq \frac{1}{(N-1)^2} \sum_{\substack{m=1 \\ m \neq i}}^{N} 1 + \frac{1}{(N-1)^2(N-2)} \Big( \sum_{\substack{m=1 \\ m \neq i}}^{N} 1 \Big)^2.
\end{aligned}
$$

The last inequality is by $\forall m, \ |g_{im}| \leq 1$, and the direct calculation implies that

$$
\left| \frac{S_{g_i}^2}{N-1} \right| \leq \frac{1}{N-1} + \frac{1}{N-2} \leq \frac{2}{N-2}.
$$

$\square$

**Lemma 3.3** $\mathbb{E}_{\boldsymbol{Z_k}, \boldsymbol{Y_k}}[\bar{g}_i^{\,2} \bar{g}_j^{\,2}] = \mathbb{E}_{\boldsymbol{Z_k}, \boldsymbol{Y_k}}[\bar{g_1}^2 \bar{g_2}^2] \ \forall i, j \text{ where } i \neq j.$

*Proof.*

$$
\begin{aligned}
\sum_{\substack{m,n=1 \\ \{m,n\} \neq \{i,j\}}}^{N} g_{im} g_{jn} &= \sum_{\substack{m,n=1 \\ \{m,n\} \neq \{i,j\}, \, m \neq n}}^{N} g_{im} g_{jn} + \sum_{\substack{m,n=1 \\ n \neq \{i,j\}}}^{N} g_{ij} g_{jn} + \sum_{\substack{m,n=1 \\ m \neq \{i,j\}}}^{N} g_{im} g_{ji} \\
&\quad + \sum_{\substack{m,n=1 \\ m \neq \{i,j\}}}^{N} g_{im} g_{jm} + g_{ij} g_{ji} \quad \text{(since these terms are disjoint terms)} \\
&\overset{d}{=} \sum_{\substack{m',n'=1 \\ \{m',n'\} \neq \{1,2\}, \, m' \neq n'}}^{N} g_{1m'} g_{2n'} + \sum_{\substack{m',n'=1 \\ n' \neq \{1,2\}}}^{N} g_{12} g_{2n'} + \sum_{\substack{m',n'=1 \\ m' \neq \{1,2\}}}^{N} g_{1m'} g_{21} \\
&\quad + \sum_{\substack{m',n'=1 \\ m' \neq \{1,2\}}}^{N} g_{1m'} g_{2m'} + g_{12} g_{21} \\
&= \sum_{\substack{m',n'=1 \\ m' \neq 1, n' \neq 2}}^{N} g_{1m'} g_{2n'}.
\end{aligned}
$$

Proof of claim above:
As the summation can be divided into five distinct cases,

Let $i$ and $j$ be arbitrary such that $i \neq j$,

Case 1: $\{m, n\} \neq \{i, j\}$ and $m \neq n$,

$$
\begin{aligned}
g_{im}g_{jn} &= I(Z_{k,m} \leq Z_{k,i})[I(Y_{k,m} = 1) - Z_{k,m}]I(Z_{k,n} \leq Z_{k,j})[I(Y_{k,n} = 1) - Z_{k,n}] \\
&= h_1(Z_{k,i}, Z_{k,m}, Z_{k,j}, Z_{k,n}, Y_{k,m}, Y_{k,n}) \\
&\overset{d}{=} h_1(Z_{k,1}, Z_{k,m'}, Z_{k,2}, Z_{k,n'}, Y_{k,m'}, Y_{k,n'}). \qquad \text{where } \{m', n'\} \neq \{1, 2\} \text{ and } m' \neq n'
\end{aligned}
$$

Case 2: $m = j$ and $n \neq \{i, j\}$,

$$
\begin{aligned}
g_{ij}g_{jn} &= I(Z_{k,j} \leq Z_{k,i})[I(Y_{k,j} = 1) - Z_{k,j}]I(Z_{k,n} \leq Z_{k,j})[I(Y_{k,n} = 1) - Z_{k,n}] \\
&= h_2(Z_{k,i}, Z_{k,j}, Z_{k,n}, Y_{k,j}, Y_{k,n}) \\
&\overset{d}{=} h_2(Z_{k,1}, Z_{k,2}, Z_{k,n'}, Y_{k,2}, Y_{k,n'}). \qquad \text{where } m' = 2 \text{ and } n' \neq \{1, 2\}
\end{aligned}
$$

Case 3: $n = i$ and $m \neq \{i, j\}$,

$$
\begin{aligned}
g_{im}g_{ji} &= I(z_{k,m} \leq Z_{k,i})[I(Y_{k,m} = 1) - Z_{k,m}]I(Z_{k,i} \leq Z_{k,j})[I(Y_{k,i} = 1) - Z_{k,i}] \\
&= h_3(Z_{k,i}, Z_{k,m}, Z_{k,j}, Y_{k,m}, Y_{k,i}) \\
&\overset{d}{=} h_3(Z_{k,1}, Z_{k,m'}, Z_{k,2}, Y_{k,m'}, Y_{k,1}). \qquad \text{where } n' = 1 \text{ and } m' \neq \{1, 2\}
\end{aligned}
$$

Case 4: $m = n \neq \{i, j\}$,

$$
\begin{aligned}
g_{im}g_{jm} &= I(Z_{k,m} \leq Z_{k,i})[I(Y_{k,m} = 1) - Z_{k,m}]I(Z_{k,m} \leq Z_{k,j})[I(Y_{k,m} = 1) - Z_{k,m}] \\
&= h_4(Z_{k,i}, Z_{k,m}, Z_{k,j}, Y_{k,m}) \\
&\overset{d}{=} h_4(Z_{k,1}, Z_{k,m'}, Z_{k,2}, Y_{k,m'}). \qquad \text{where } m' = n' \neq \{1, 2\}
\end{aligned}
$$

Case 5: $m = j$ and $n = i$,

$$
\begin{aligned}
g_{ij}g_{ji} &= I(Z_{k,j} \leq Z_{k,i})[I(Y_{k,j} = 1) - Z_{k,j}]I(Z_{k,i} \leq Z_{k,j})[I(Y_{k,i} = 1) - Z_{k,i} \\
&= h_5(Z_{k,i}, Z_{k,j}, Y_{k,i}, Y_{k,j}) \\
&\overset{d}{=} h_5(Z_{k,1}, Z_{k2}, Y_{k,1}, Y_{k,2}). \qquad \text{where } m' = 2 \text{ and } n' = 1
\end{aligned}
$$

This follows from the fact that $Z_{k,i}$'s and $Y_{k,i}$'s are i.i.d. random variables. Therefore,

$$
(\sum_{\substack{m,n=1 \\ m \neq i, n \neq j}}^{N} g_{im}g_{jn})^2 \overset{d}{=} (\sum_{\substack{m',n'=1 \\ m' \neq 1, n' \neq 2}}^{N} g_{1m'}g_{2n'})^2
$$

$$
\bar{g_i}^2 \bar{g_j}^2 \overset{d}{=} \bar{g_1}^2 \bar{g_2}^2
$$

$$
\mathbb{E}_{\boldsymbol{Z_k}, \boldsymbol{Y_k}}[\bar{g_i}^2 \bar{g_j}^2] = \mathbb{E}_{\boldsymbol{Z_k}, \boldsymbol{Y_k}}[\bar{g_1}^2 \bar{g_2}^2].
$$

$\square$

**Lemma 3.4** $\lim_{N \to \infty} \mathbb{E}_{\boldsymbol{Z_k}, \boldsymbol{Y_k}}[\bar{g_1}^2 \bar{g_2}^2] = \mathbb{E}^2_{Z_k'}[d_k^2(Z_k')]$.

*Proof.* For arbitrary $i$, $\bar{g_i} \overset{\mathbb{P}}{\to} \mu_i$ by the strong law of large number. Thus, $\bar{g_1} \overset{\mathbb{P}}{\to} \mu_1$ and $\bar{g_2} \overset{\mathbb{P}}{\to} \mu_2$. Therefore, since marginal convergence in probability implies joint convergence in probability (Vaart, 1998),

$$
(\bar{g_1}, \bar{g_2}) \overset{\mathbb{P}}{\to} (\mu_1, \mu_2).
$$

By continuous mapping theorem, since $f(x, y) = x^2 y^2$ is a continuous function,

$$
\bar{g_1}^2 \bar{g_2}^2 \overset{\mathbb{P}}{\to} \mu_1^2 \mu_2^2.
$$

Additionally, since $|\bar{g_1}^2 \bar{g_2}^2| \leq 1 \; \forall N$, it is uniformly bounded and thus uniformly integrable. Combining this with the fact that it converges in probability,

$$
\lim_{N \to \infty} \mathbb{E}_{\boldsymbol{Z_k}, \boldsymbol{Y_k}}[\bar{g_1}^2 \bar{g_2}^2] = \mathbb{E}_{\boldsymbol{Z_k}, \boldsymbol{Y_k}}[\mu_1^2 \mu_2^2].
$$

Thus,

$$\lim_{N \to \infty} \mathbb{E}_{\boldsymbol{Z_k}, \boldsymbol{Y_k}}[\bar{g_1}^2 \bar{g_2}^2] = \mathbb{E}_{\boldsymbol{Z_k}, \boldsymbol{Y_k}}[\mu_1^2 \mu_2^2] = \mathbb{E}_{\boldsymbol{Z_k}, \boldsymbol{Y_k}}[\mu_1^2] E_{\boldsymbol{Z_k}, \boldsymbol{Y_k}}[\mu_2^2] = \mathbb{E}_{Z_k'}^2[d_k^2(Z_k')].$$

$\square$

## C EXPECTED SQUARED DIFFERENCE (ESD) PSEUDOCODE

In this section, we provide a pseudocode for calculating the Expected Squared Difference (ESD) for a given batch output.

**Algorithm 1:** Pytorch-like Pseudocode: Expected Squared Difference (ESD)

```
# n:number of samples in a mini-batch.
# confidence:1D tensor of n elements containing max softmax
 outputs of each sample from a neural network.
# correct:1D tensor of n elements containing 1/0 corresponding
 to the correctness of each sample.
def ESD_loss(n, confidence, correct):
  # compute the difference between confidence and correctness.
  diff = correct.float() - confidence

  # Prepare the split between inner and outer expectation
   estimation.
  split = torch.ones(n,n) - torch.eye(n)

  # compute the inner expectation estimation.
  confidence_mat= confidence.expand(n,n)
  ineq = torch.le(confidence_mat,confidence_mat.T).float()
  diff_mat = diff.view(1,n).expand(n,n)
  x_mat = torch.mul(diff_mat,ineq)*split
  mean_row = torch.sum(x_mat, dim = 1)/(n-1)
  x_mat_squared = torch.mul(x_mat, x_mat)
  var = 1/(n-2) * torch.sum(x_mat_squared,dim=1) - (n-1)/(n-2) *
   torch.mul(mean_row,mean_row)

  # compute the outer expectation estimation.
  d_k_sq_vector = torch.mul(mean_row, mean_row) - var/(n-1)
  ESD = torch.sum(d_k_sq_vector)/n
  return ESD
```

## D UNBIASED ESTIMATORS AND ITS GRADIENT

Since ESD is an unbiased estimator,

$$\mathbb{E}_{\boldsymbol{Z_k}, \boldsymbol{Y_k}}[ESD] = \mathbb{E}_{Z_k'}[d_k^2(Z_k')]$$

$$\nabla_\theta \mathbb{E}_{Z_k'}[d_k^2(Z_k')] = \nabla_\theta \mathbb{E}_{\boldsymbol{Z_k}, \boldsymbol{Y_k}}[ESD]$$

$$= \nabla_\theta \mathbb{E}_{\boldsymbol{X}, \boldsymbol{Y}}[ESD] \quad \text{(by law of unconscious statistician)}$$

$$= \mathbb{E}_{\boldsymbol{X}, \boldsymbol{Y}}[\nabla_\theta ESD]. \quad \text{(since } (\boldsymbol{X}, \boldsymbol{Y}) \text{ are independent of } \theta)$$

Thus, the gradient of the unbiased estimator is on average the gradient of the desired metric.

## E ADDITIONAL INFORMATION ON THE CHOICE OF LAMBDA RANGE

To validate the choice of $\lambda$ grid range in our experimental settings, we plot the variations in accuracy with respect to increasing $\lambda$ for all methods (Figure 5). We see that our choice of $\lambda$ contains points

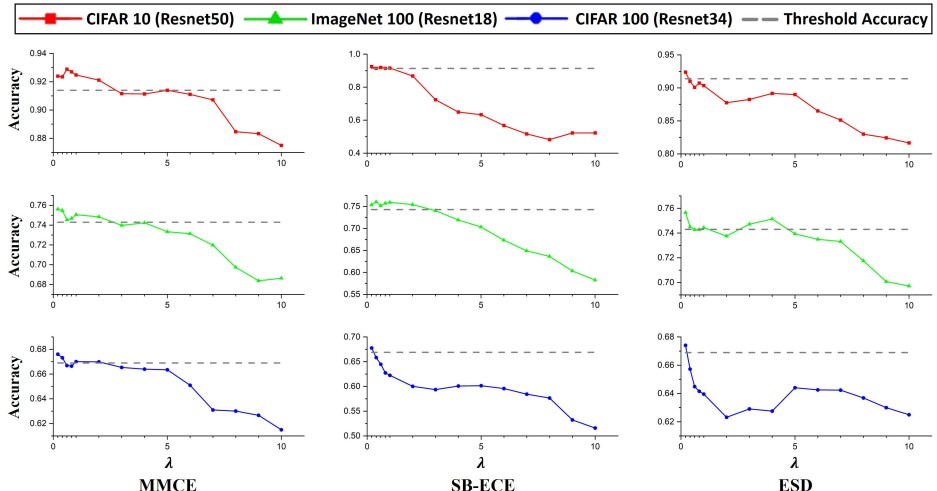

Figure 5: Accuracy plot with respect to varying values of $\lambda$ across different datasets and models trained with MMCE, SB-ECE, and ESD. The threshold accuracy represents the value 1.5% below the baseline accuracy, which was used as the model selection criterion as stated in section 5.2.

within the acceptable range of accuracy (i.e., within 1.5% degradation in accuracy compared to baseline). However, after a particular $\lambda$ value, which is different on a per method per dataset basis, the accuracy decreases consistently with increasing $\lambda$. As such, the grid chosen is suitable for the experiments conducted.

# F    TRAINABLE CALIBRATION MEASURES UNDER DISTRIBUTION SHIFT

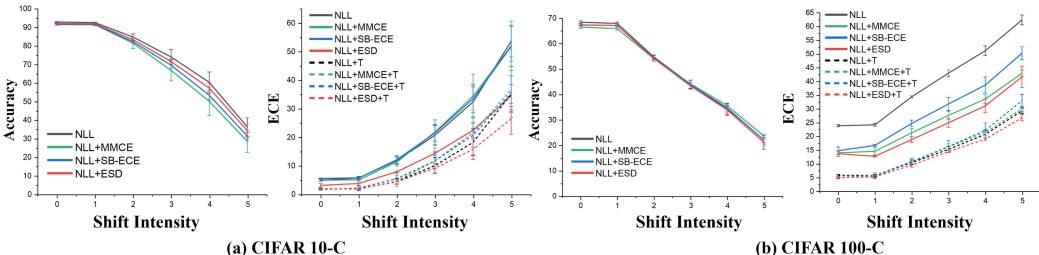

Figure 6: Accuracy and ECE plot for varying intensities of distribution shifts in (a) CIFAR 10-C and (b) CIFAR 100-C across models trained using NLL as well as those jointly trained with an auxiliary calibration objective (i.e., NLL+MMCE, NLL+SB-ECE, NLL+ESD) and their performance after post-processing with temperature scaling (i.e., NLL+T, NLL+MMCE+T, NLL+SB-ECE+T, NLL+ESD+T).

In Figure 6, we evaluate the performance of calibration methods under distribution shift benchmark datasets, CIFAR 10-C and CIFAR 100-C, introduced in Hendrycks & Dietterich (2019). For models jointly trained with an auxiliary calibration objective (NLL+MMCE, NLL+SB-ECE, NLL+ESD), we observe that they are more robust to distribution shifts when compared to those solely trained with NLL. In addition, prior work (Ovadia et al., 2019), has shown that the calibration performance of a neural network after temperature scaling may be significantly reduced under distribution shifts. Our results on CIFAR 10-C and CIFAR 100-C imply a similar trend. Stacking calibration during training methods with temperature scaling (NLL+MMCE+T, NLL+SB-ECE+T, NLL+ESD+T), we observe that they perform comparable to temperature scaled models trained with NLL (NLL+T) with the exception of ESD, where it performs marginally better. As such, training with ESD could potentially improve a model's robustness after temperature scaling regarding distribution shifts and mitigate this issue.

