# OpenReview forum: "ESD: Expected Squared Difference as a Tuning-Free Trainable Calibration Measure"
_ICLR.cc/2023/Conference — ICLR 2023 poster_

### Official Review · Reviewer_ywx4 · 2022-10-24

**Confidence:** 4
**Correctness:** 4
**Technical Novelty And Significance:** 4
**Empirical Novelty And Significance:** 4
**Recommendation:** 8

**Clarity, Quality, Novelty And Reproducibility:**

All is novel and reproducible and clear, except for the weaknesses highlighted above.

**Strength And Weaknesses:**

Strengths:
* The paper is well-written and clear.
* The proposed method ESD has been justified theoretically and the ESD estimator has been proved to be unbiased and consistent.
* The experiments on 5 datasets demonstrate that ESD results in better calibrated probabilities than earlier methods, while not sacrificing more accuracy.


Weaknesses:
* The intuition behind the proposed method could have been explained more, perhaps on a toy example.

* More information about the experimental results could have been reported, for example as an appendix. It would be good to know whether the range of values for the hyperparameter lambda is sufficient and which values were eventually chosen by the method on different datasets. Also, it would have been good to report NLL as an evaluation measure, in addition to accuracy and ECE. E.g., Table 1 could have had extra rows for NLL, NLL after TS, and NLL after VS, or alternatively, this information could have been provided in the appendix. The paper does not mention why accuracy after vector scaling is so low for CIFAR100 and ANLI. Can there really be so much overfitting even though the number of classes is quite high?

* The conclusion fails to mention that some accuracy has been sacrificed to achieve better calibration. On the positive side, this fact has been mentioned in Section 6 about experimental results.

* Notation $k$ is overloaded in Section 3 - first it is the total number of classes, then just any class in Eq.(1), and then the class with maximum output probability in Eq.(2). This can create confusion, in particular because Eq.(1) is about being classwise-calibrated whereas later the paper only studies confidence calibration.

Minor weaknesses:
* Table 3 and Section 7.3 fails to mention which dataset it is about. I assume CIFAR100?

* The last sentence of page 7 (first paragraph of Section 6) states the decrease of accuracy by more than 2% - is it not about 1% in this case (47.08 vs 48.0) because the comparison is all between ESD and ESD+VS?

**Summary Of The Paper:**

The paper proposes a new method to improve calibration of multi-class classifiers. It is similar to the existing methods of MMCE and SB-ECE in the sense that it adds a trainable auxiliary loss function (Expected Squared Difference, ESD) to the cross-entropy loss (NLL) during training of the classifier. Similarly to SB-ECE, the paper uses interleaved training where the new auxiliary loss and NLL are trained on separate parts of the training set. ESD is parameter-free and its estimator is unbiased. The experiments on 5 datasets demonstrate that compared to MMCE, SB-ECE and the baseline NLL it results in better-calibrated predictions (lower ECE) both before and after post-hoc calibration with temperature scaling or vector scaling. About 1% of accuracy is sacrificed compared to the baseline NLL.

**Summary Of The Review:**

The paper proposes a parameterless method and improves the state-of-the-art. The proposed method ESD is simpler and better than the existing methods MMCE and SB-ECE. There are some weaknesses highlighted above but these can be fixed for the camera-ready v
ersion.

---

> ### Author Response · Authors · 2022-11-09
> **Response to Reviewer ywx4**
>
> We thank the reviewer for the detailed review and for their feedback. We respond to their comments below:
>
> Q1. The intuition behind the proposed method could have been explained more, perhaps on a toy example.
>
> A1. Thank you for bringing this to our attention! We agree that it is important to have toy examples to help better explain the intuition of the method. The toy example will involve two graphs (one for a calibrated and the other for an uncalibrated network) to depict the calibration error as the area of the difference between the two functions corresponding to the two expectations in Eq. 7 in our paper. This toy example will be included in the updated manuscript which will be hopefully published  by this weekend as mentioned in the general response.
>
> Q2. More information about the experimental results could have been reported, for example as an appendix. It would be good to know whether the range of values for the hyperparameter lambda is sufficient and which values were eventually chosen by the method on different datasets. Also, it would have been good to report NLL as an evaluation measure, in addition to accuracy and ECE. E.g., Table 1 could have had extra rows for NLL, NLL after TS, and NLL after VS, or alternatively, this information could have been provided in the appendix. The paper does not mention why accuracy after vector scaling is so low for CIFAR100 and ANLI. Can there really be so much overfitting even though the number of classes is quite high?
>
> A2. Thank you very much for the suggestion. We will add a subsection in the appendix with a mini experiment to evaluate the choice of range of lambda value in the revised version. With regards to the accuracy drop for vector scaling on CIFAR100 and ANLI, while we don’t have a detailed response on this as of yet, we plan to analyze the reliability plots as well as the training curve to gleam an explanation on this phenomena.
>
> Q3. The conclusion fails to mention that some accuracy has been sacrificed to achieve better calibration. On the positive side, this fact has been mentioned in Section 6 about experimental results.
>
> A3. We agree with this sentiment and have thus made changes to the conclusion. This changes can be seen in the revised version that will soon be published as mentioned in the general response.
>
> Q4. Notation k is overloaded in Section 3 - first it is the total number of classes, then just any class in Eq.(1), and then the class with maximum output probability in Eq.(2). This can create confusion, in particular because Eq.(1) is about being classwise-calibrated whereas later the paper only studies confidence calibration.
>
> A4. Thank you for alerting us of this! We agree that the changing definition of what k represents can be confusing to the reader. As such, we have used different notation for different cases.
>
> Q5. Table 3 and Section 7.3 fails to mention which dataset it is about. I assume CIFAR100?
>
> A5. Thank you for bringing this to our attention! Yes the dataset used in section 7.3, Table 3 was CIFAR100 and we have made changes relating to that section in the revised draft.
>
> Q6. The last sentence of page 7 (first paragraph of Section 6) states the decrease of accuracy by more than 2% - is it not about 1% in this case (47.08 vs 48.0) because the comparison is all between ESD and ESD+VS?
>
> A6. Thank you very much for the question. The more than 2% decrease was when compared to baseline. We have changed the wording in the revised version to avoid such confusion for the readers.

---

> > ### Author Response · Authors · 2022-11-18
> > **[Update] Response to Reviewer ywx4**
> >
> > We greatly appreciate the reviewer for pointing out the concern of accuracy drop after vector scaling for CIFAR100 and ANLI. For all the vector scaling experiments, we have been using the LBFGS optimizer with a Strong Wolfe line search function (https://pytorch.org/docs/stable/generated/torch.optim.LBFGS.html) with lr parameter set to 0.01. After further analysis of this issue, for CIFAR100 baseline and ANLI baseline (where the accuracy drop was above 1.5%), we noticed that the performance of vector scaling using Strong Wolfe line search function could be sensitive to the aforementioned lr parameter. In particular, lowering the lr parameter reduced the drop in accuracy while giving similar ECE values obtained when using the previously set lr (lr = 0.01). As such, we have rerun the vector scaling experiments with smaller lr (lr = 0.0001) for those with an accuracy drop of more than 1.5 %  after vector scaling and have updated the manuscript accordingly.
> >
> > Additionally, taking into account of the reviewers' feedback and suggestions, we have uploaded a revised manuscript with the changes mentioned in General Response. We would greatly appreciate if you could read through the revised manuscript.

---

### Official Review · Reviewer_smEk · 2022-10-24

**Confidence:** 4
**Correctness:** 4
**Technical Novelty And Significance:** 3
**Empirical Novelty And Significance:** 3
**Recommendation:** 8

**Clarity, Quality, Novelty And Reproducibility:**

Clarity&Quality: this paper is well-written.
Novelty: notable novelty
reproducibility: Settings and parameters are listed and the authors committed to release the code publicly

**Strength And Weaknesses:**

Strengths:
1. Build a well-established  trainable calibration objective loss ESD, which is binning-free and does not need hyperparameters.
2. Extensive experiments to prove that the esd produces more accurate ECE without tuning the hyperparameters.
Weaknesses:
1. This calibration method itself does not need any hyper-parameter but still incorporates some other hyperparameter, such as the regularizer hyperparameter λ, monte carlo sample number N, etc.

**Summary Of The Paper:**

This paper present Expected Squared Difference (ESD), a tuning-free (i.e., hyperparameter-free) trainable calibration objective loss, which views the calibration error fromthe perspective of the squared difference between two expectations.

**Summary Of The Review:**

This paper proposes a hyperparameter-free calibration method, ESD and produces accurate predictions and calibration results.

---

> ### Author Response · Authors · 2022-11-09
> **Response to Reviewer smEk**
>
> We thank the reviewer for their feedback. We respond to their comments below:
>
> Q1. This calibration method itself does not need any hyper-parameter but still incorporates some other hyperparameter, such as the regularizer hyperparameter λ, monte carlo sample number N, etc.
>
> A1. We agree that there are still hyperparameters such as regularizer hyperparameter and monte carlo sample number N, etc., however the focus of this paper is primarily about achieving a trainable tuning-free metric. As with most regularizers, we jointly optimize ESD with NLL and thus cannot avoid the need for a balancing hyperparameter $\lambda$ between the two. In spite of this, the lack of internal hyperparameter in ESD has been shown to be effective in reducing training time compared to competing baselines as shown in ablation study section 7.1.

---

> > ### Author Response · Authors · 2022-11-19
> > **[Update] Response to Reviewer smEK**
> >
> > We appreciate the reviewer again for their valuable feedback. Taking into account of the reviewers' suggestions, we have uploaded a revised manuscript with the changes stated in the General Response. We would greatly appreciate if you could read through the revised manuscript.

---

> > > ### Comment · Reviewer_smEk · 2022-12-06
> > > **thanks for the reponse**
> > >
> > > Thanks for the reponse and the revision. I will raise my score accordingly.

---

### Official Review · Reviewer_XMyj · 2022-10-25

**Confidence:** 4
**Correctness:** 4
**Technical Novelty And Significance:** 3
**Empirical Novelty And Significance:** 3
**Recommendation:** 6

**Clarity, Quality, Novelty And Reproducibility:**

Clarity: The paper is well written.
Quality: Empirical and theoretically solid.
Novelty: The proposed method of calibration is novel especially in terms of its application at train time.
Reproducibility: The authors have promised to release code publicly.

**Strength And Weaknesses:**

Strengths:
- The paper is easy to follow and well-written.
- There has been a lot of research focused on post-hoc calibration methods in the literature but this paper looks upon the solving the mis-calibration at the training time without significant additional computational costs. This problem is particularly challenging as the overfitting is a significant challenge to solve this task.
- The paper achieves impressive results on multiple benchmarks of calibration while proposing a theoretically concrete approach to calibration.

Weaknesses:
- Since this is a training based calibration method, I would suggest the authors add calibration results on large scale datasets such as ImageNet.
- An important addition to the paper, should be along the lines of calibration performance under distribution shifts [a] as this has been known to be difficult for post-hoc calibration methods but I would assume the proposed method should be able to perform better under such a scenario.

References:
- Ovadia et al. Can You Trust Your Model’s Uncertainty? Evaluating Predictive Uncertainty Under Dataset Shift. NeurIPS 2019.


**Summary Of The Paper:**

The paper introduces a hyperparameter free training based calibration method using proposed calibration metric, namely expected squared difference (ESD). The proposed calibration metric is inspired from the recently introduced KS metric that compares cumulative distribution.

**Summary Of The Review:**

I am in favor of acceptance of the paper as its a novel way of calibration and proposed method looks clean. I suggest some few additions to the paper which will make it stronger. If my concerns are addressed, I will increase my rating.

---

> ### Author Response · Authors · 2022-11-09
> **Response to Reviewer XMyj**
>
> We thank the reviewer for a very thorough and detailed review. We respond to individual comments below:
>
> Q1. Since this is a training based calibration method, I would suggest the authors add calibration results on large scale datasets such as ImageNet.
>
> A1. We appreciate your suggestion. We are currently running a larger scale vision experiment (Imagenet-100 on ResNet18) and as mentioned in the general response, we hope to publicize the results in the revised manuscript by this weekend.
>
> Q2. An important addition to the paper, should be along the lines of calibration performance under distribution shifts [a] as this has been known to be difficult for post-hoc calibration methods but I would assume the proposed method should be able to perform better under such a scenario.
>
> A2. Thank you for the comment. We fully agree that it would be interesting to show the calibration performance under distribution shift. As such, we have started running the related experiments and hope to publish them in the revised version.

---

> > ### Author Response · Authors · 2022-11-17
> > **[Update] Response to Reviewer XMyj**
> >
> > We thank the reviewer again for their valuable suggestions. We have taken into consideration the suggestions and have added sections relating to them in the current revised manuscript (Imagenet result in Table1 and data shift analysis in Appendix F). The full list of changes made could be seen in the General Response. We would greatly appreciate if you could read through the revised manuscript.

---

> > > ### Comment · Reviewer_XMyj · 2022-11-22
> > > **Response to Rebuttal**
> > >
> > > Thanks for addressing my concerns. I am overall satisfied with the authors response to my feedback and thus will increase my score to 7.

---

### Official Review · Reviewer_md7e · 2022-10-25

**Confidence:** 3
**Correctness:** 4
**Technical Novelty And Significance:** 3
**Empirical Novelty And Significance:** 2
**Recommendation:** 6

**Clarity, Quality, Novelty And Reproducibility:**

- Clarity: overall this article is well-organized and easy to follow.
- Quality and novelty: the paper is solid and the novelty is intermediate.
- Reproducibility: The authors promise to release  the code.

**Strength And Weaknesses:**

#### Strengths
- Overall the paper is well written and easy to follow.
- The idea of ESD is very interesting and should be able to bring a positive impact to the AI fields not limited to both vision and NLP domain.
- Strong theoretic explanation with solid mathematical functions.

#### Weaknesses
- The accuracy has no obvious advantages when compared to other competing baselines, according to Table 1.
- As the paper proposes a general calibration loss, I would like to suggest the authors to conduct the experiments on the more general large-scale vision datasets (e.g., MS-COCO and ImageNet).

**Summary Of The Paper:**

The paper proposes a trainable objective less ESD as the extra calibration loss jointly optimized with the NLL loss during training. It is claimed as a binning-free objective lease without need to tune any additional parameters. The experiments were extensively conducted across three architectures (MLPs, CNNs, and Transformers) and both vision and NLP datasets. The experimental results seem to strongly validate the efficacy of the proposed method.

**Summary Of The Review:**

Based on the above statements, I would like to weakly accept the paper at the initial stage. After carefully reading the other reviewers' comments and the authors' response, I would like to keep the initial rating.

---

> ### Author Response · Authors · 2022-11-09
> **Response to Reviewer md7e**
>
> We thank the reviewer for reviewing our work and for the feedback and questions. We address their concerns below:
>
>
> Q1. The accuracy has no obvious advantages when compared to other competing baselines, according to Table 1.
>
> A1. We agree with this sentiment. However, we feel that it is also important to note that the research of calibration is in general not particularly concerned on the improvement of accuracy but rather improving calibration without sacrificing much accuracy. In that regard, our method was able to achieve the lowest ECE with not more than 1.5% degradation of accuracy when compared to the baseline.
>
> Q2. As the paper proposes a general calibration loss, I would like to suggest the authors to conduct the experiments on the more general large-scale vision datasets (e.g., MS-COCO and ImageNet).
>
> A2. Thank you for the suggestion! We have recently started running a larger scale vision experiments (Imagenet-100 with ResNet18). As mentioned in the general response, we hope to publish the results in the updated manuscript by this weekend.

---

> > ### Author Response · Authors · 2022-11-19
> > **[Update] Response to Reviewer md7e**
> >
> > We thank the reviewer again for their valuable suggestions. We have taken into consideration the suggestions and have added sections relating to them in the current revised manuscript (Imagenet result in Table1). The complete list of changes that were made can be seen in the General Response. We would greatly appreciate if you could read through the revised manuscript.

---

### Author Response · Authors · 2022-11-09
**General Response to the Reviewers**

We would like to thank the reviewers for the detailed review and feedback and addressed each noted question separately. With regards to the additional experiments requested by the reviewers, the experiments we are currently conducting are:

1. Seeing the performance comparison of ESD and competing baselines on ImageNet-100 (with ResNet18).

2. Observing the calibration performance during the distribution shift (specifically using the benchmarking dataset introduced in the following paper - Dan Hendrycks et al. Benchmarking Neural Network Robustness to Common Corruptions and Perturbations. ICLR2019).

As such, the revised manuscript will be posted after the completion of the above experiments, hopefully by this weekend. We thank the reviewers in advance for their patience.

---

> ### Author Response · Authors · 2022-11-13
> **[Update] Revised Manuscript Uploaded and Comments**
>
> We thank the reviewers again for their patience. We have just uploaded a revised version with the following changes:
>
> [Major changes]
>
> 1. Toy Example showing the intuition behind the proposed ESD method. We have included this in Appendix A.
> 2. Larger scale vision experimental result on Imagenet-100 has been added to Table 1. To fit the result into Table 1, we have swapped the row and column of Table 1.
> 3. We have added a section in Appendix E to verify whether the range of values for the hyperparameter lambda in the grid is suitable.
> 4. We have also included a pseudocode for ESD in Appendix C.
>
> [Minor changes]
>
> We thank the reviewers for the valuable suggestions to clarify certain parts of the paper that may confuse potential readers. We have taken these suggestions into account and have changed the wording in those areas.
>
> [Comments]
>
> We are yet to finish analyzing the data shift experiment suggested, and find a suitable explanation for the accuracy drop observed for the CIFAR100 baseline and ANLI baseline after vector scaling. We hope to find a satisfying response to the reviewers regarding these issues within a few days. We thank the reviewers in advance again for their patience.

---

> ### Author Response · Authors · 2022-11-17
> **[Update #2] Revised Manuscript**
>
> We thank the reviewers again for their patience during this rebuttal period. We have uploaded a second revised version of the manuscript with the following additions:
>
> * Analysis on calibration metrics under distribution shifts (Appendix F).
>
> * In Table 1, we updated the accuracy and ECE values after vector scaling for CIFAR100-baseline and ANLI-baseline using a lower learning rate during vector scaling. For further information on this, please refer to the comment made to Reviewer ywx4.

---

### Decision · Program_Chairs · 2023-01-20

**Decision:**

Accept: poster

**Justification For Why Not Higher Score:**

Trainable calibration objectives is a somewhat incremental space since works have already been proposed in this area, especially SB-ECE. The improvements are statistically significant on the limited set of experiments shown, but haven't been checked on even ImageNet scale which seems standard practice particularly for these models. The baseline accuracy #s across the datasets aren't strong enough to be convincing enough that even in the space of trainable calibration methods, that this is _the_ best method. A true proposal for the best calibration would further compare to the broader space of methods improving calibration (e.g., efficient ensembles).

**Justification For Why Not Lower Score:**

Reviewers have uniform agreement of accept, and the method is simple and interesting to consider.

**Metareview: Summary, Strengths And Weaknesses:**

Uncertainty can be challenging in deep learning, and this work aims to improve their calibration. In particular, the authors propose a calibration objective Expected Squared Difference that is differentiable and tuning-free, able to be used during training. It is similar to recent works in this space such as MMCE and SB-ECE. The argument for the new proposal is that it's simpler and shows better results in practice. They validate their proposal on MNIST, CIFAR-10, and SNLI.

Reviewers found the work well-written and easy to follow. The theoretical justification was also highlighted a positive. I think the problem is quite challenging as it's easy to overfit when training for calibration specifically (see also the paper's Section 3.2.2). They build on Karandikar et al.'s approach of interleaved training and the experiments show that among the datasets analyzed, ESD is quite better than the 2 previous methods.

All reviewers leaned toward accept and I agree with their consensus.

**Note From Pc:**

if the above contains the word "oral" or "spotlight" please see: "oral" presentation means -> notable-top-5% and "spotlight" means -> notable-top-25%. As stated in our emails, we are disassociating presentation type from AC recommendations